# Iatrogenic Incidents in Primary Molar Pulpectomy: A Case Series Report and Literature Review

Yeo Won Lim [1], Yong Kwon Chae [2], Ko Eun Lee [2], Ok Hyung Nam [3], Hyo-Seol Lee [3], Sung Chul Choi [3] and Mi Sun Kim [4],*

[1] Department of Pediatric Dentistry, Kyung Hee University Dental Hospital at Gangdong, Seoul 05278, Republic of Korea; dla0182@naver.com
[2] Department of Pediatric Dentistry, Kyung Hee University Medical Center, Seoul 02447, Republic of Korea; van_0418@naver.com (Y.K.C.); olivedlr@naver.com (K.E.L.)
[3] Department of Pediatric Dentistry, Kyung Hee University College of Dentistry, Kyung Hee University Medical Center, Seoul 02447, Republic of Korea; khupedo2012@naver.com (O.H.N.); snowlee@khu.ac.kr (H.-S.L.); pedochoi@khu.ac.kr (S.C.C.)
[4] Department of Pediatric Dentistry, Kyung Hee University College of Dentistry, Kyung Hee University Hospital at Gangdong, Seoul 05278, Republic of Korea
* Correspondence: happystation@empal.com

**Abstract:** Pulpectomy is a common treatment for severe carious lesions in primary molars. Care should be taken during pulpectomy of the primary teeth for successive permanent teeth. This case series report aimed to describe the cases of three patients who experienced excessive extrusion of canal filling materials and file separation during primary molar pulpectomy. The presence of separated files or excessive overfilling materials observed around successive permanent tooth germs leads to underdevelopment and may trigger cystic changes owing to abscess formation. The most important aspect is to preserve the unerupted successive developing permanent tooth. Therefore, clinicians should consider the anatomy of the primary molars prior to endodontic treatment, be careful when manipulating instruments during pulpectomy, provide appropriate treatment if iatrogenic incidents occur during endodontic treatment, and perform close follow-up to ensure the successful development and eruption of subsequent permanent teeth.

**Keywords:** pulpectomy; overfilling; file separation; primary teeth; marsupialization; underdevelopment





## 1. Introduction

Pulpectomy of the primary teeth is a conservative treatment approach for severe caries, chronic inflammation, or pulp necrosis. The fundamental objectives of pulp therapy in the primary dentition are to maintain the tooth in a pathologically healthy condition, preserve its function as an essential part of the primary dentition, and eliminate infection and chronic inflammation while providing relief from the pain caused by inflamed pulp. Thus, the affected tooth retains its functional status until its natural exfoliation [1–4].

However, endodontic procedures on primary and permanent teeth are frequently complicated, with a variety of factors influencing the degree of difficulty and risks associated with the choice of treatment [5]. The complexity of the root canal system and its resorption pattern in primary teeth may interfere with optimal canal filling [2,6]. As a result, there can be a high incidence of adverse events during primary molar endodontic treatment. Such accidents can occur at any stage of endodontic treatment, potentially resulting in treatment failure [3,7]. The fracture of an endodontic file or overfilling beyond the apex during pulpectomy are serious complications. The inability to retrieve these teeth may lead to abscess formation, root resorption, and foreign body reactions, which can hinder optimal preparation and obturation, leading to the failure and alteration of the eruption of

successive teeth. [8,9]. Thus, one of the requirements for an appropriate root-canal filling material for primary tooth pulpectomies is the resorption of the filling material, which has similar rates to that of the primary root [10,11]. If the material is extruded beyond the apex, it should be resorbable and nontoxic to periapical tissues and the permanent tooth germ [12]. The gold standard for root canal filling materials for primary teeth with these properties is iodoform-based root canal filling materials, such as Vitapex® (Neo Dental Chemical Products Co., Tokyo, Japan), which consist of a premix of calcium hydroxide and iodoform with the addition of silicone oil [13,14]. Despite these properties, excessive extrusion of the canal filling material can affect successive permanent teeth [2,15].

Because the root canals of primary teeth are wider and straighter than those of permanent teeth, file separation during pulpectomy in primary teeth rarely occurs; however, it can occur because of incorrect instrumentation techniques or an overuse associated with an excessive amount of torque of the instrument [3,8]. Moreover, there is a lack of studies in the literature which focus on the effect on the growth and dentition of the subsequent permanent teeth, with the occurrence of overfilling or the non-absorbance of filling materials in the upper part of the subsequent permanent teeth and file fracture in the root canal of the primary teeth.

This case series report aimed to present two cases associated with excessive extrusion of the canal filling material and one case associated with file separation in primary molars and their potential impacts on subsequent permanent teeth, along with a review of the relevant previously published literature, and to discuss the prognosis that may occur in the subsequent permanent dentition with follow-up after iatrogenic incidents in primary molar pulpectomy.

## 2. Case Report

### 2.1. Case 1

A 3-year-old girl was referred to the Department of Pediatric Dentistry of Kyung Hee University Dental Hospital at Gangdong from a local dental clinic for the treatment of periapical abscess on the left mandibular primary second molar. Her medical history was unremarkable. According to the referral from the local dental clinic, her dental history was a pulpectomy with root canal filling with Vitapex® on the affected tooth 7 months ago, and re-treatment (pulpectomy with root canal filling with Vitapex®) and temporary restoration conducted 2 days ago due to a periapical abscess. Dental examination showed a buccal sinus tract and gingival swelling in the left mandibular primary second molar with a temporary restoration state. Radiological examination revealed root resorption and a significant extrusion of the filling material (Vitapex®) from the apex, affecting the area of the successive tooth germ (Figure 1a).

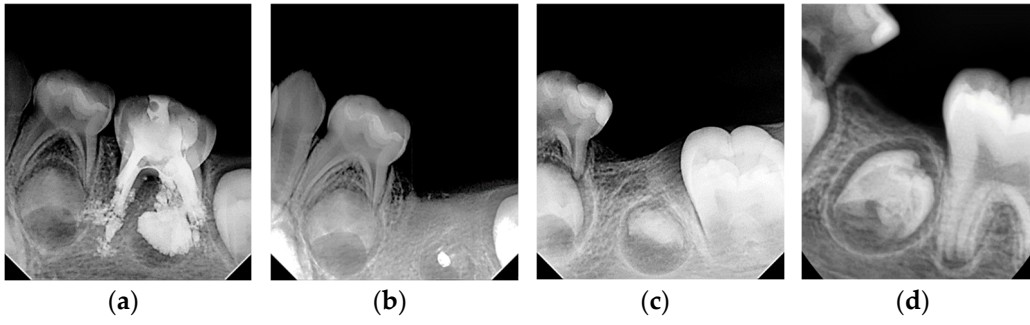

|     (a)     |     (b)     |     (c)     |     (d)     |

**Figure 1.** Periapical radiograph of left mandibular primary molar area: (**a**) initial visit; (**b**) two months after the extraction; (**c**) eleven months after the extraction; (**d**) three years after the extraction.

Under local anesthesia, the left mandibular primary second molar was extracted to prevent secondary infection and minimize damage to the successive tooth germ. The extraction site was curetted and sutured. Postoperatively, analgesics, and antibiotics were prescribed for 5 days.

Two months after extraction, most of the extruded material was resorbed, but there was a small amount left near the successive tooth germ (Figure 1b). Eleven months after extraction, radiological examination showed the underdevelopment of permanent successive tooth germ compared to the opposite side (Figures 1c and 2).

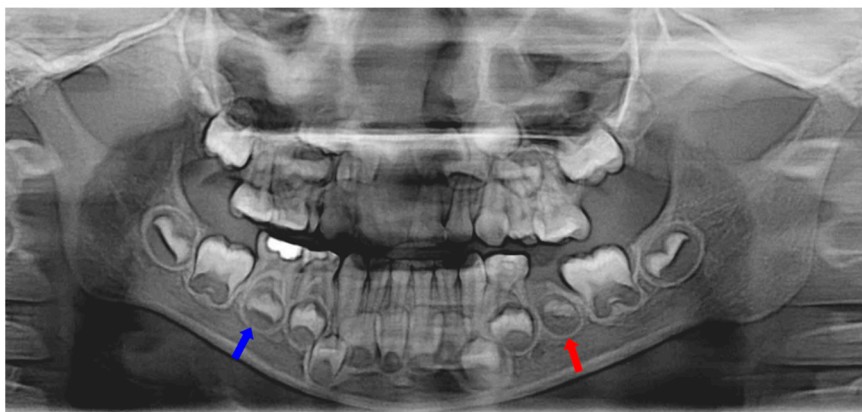

**Figure 2.** Eleven months after tooth extraction. Panoramic radiograph. The underdevelopment of the left mandibular second premolar tooth germ (red arrow) was observed compared with the opposite one (blue arrow).

A band and loop space maintainer was delivered after the eruption of the left mandibular first molar. Three years after the extraction, the development of the successive permanent tooth was observed (Figure 1d). Four years post-extraction, the successive tooth was in the pre-eruptive phase, with continuous growth and pre-eruptive movement in the alveolar bone, similar to those in the opposite tooth (Figure 3).

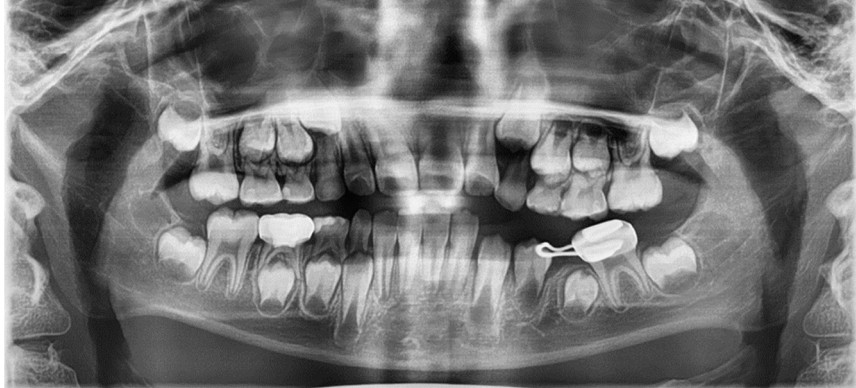

**Figure 3.** Four years after tooth extraction. Panoramic radiograph. Development of successive permanent tooth was observed (pre-eruptive phase).

### 2.2. Case 2

An 8-year-old girl was referred to the Department of Pediatric Dentistry of Kyung Hee University Dental Hospital at Gangdong from a local dental clinic for the treatment of a cystic lesion in the left mandibular primary molar area. Her medical history was unremarkable. According to the referral from the local dental clinic, her dental history was a pulpectomy with root canal filling with Vitapex® and restoration with a stainless-steel crown on the left mandibular primary molar 3 years ago. An endodontic retreatment was performed due to a recurrent periapical abscess that occurred 16 months after the initial treatment at the local clinic. Nevertheless, due to the residual periapical lesion and severe tooth mobility, the teeth were extracted one year after the retreatment at the local clinic. Radiological examination revealed a well-defined osteolytic lesion involving the premolars

and radiopaque overfilling materials (Vitapex®) around the involved premolars (Figure 4a). Cone-beam computed tomographic (CBCT) view, cortical bone thinning, and expansion of the lesion were observed (Figure 4b).

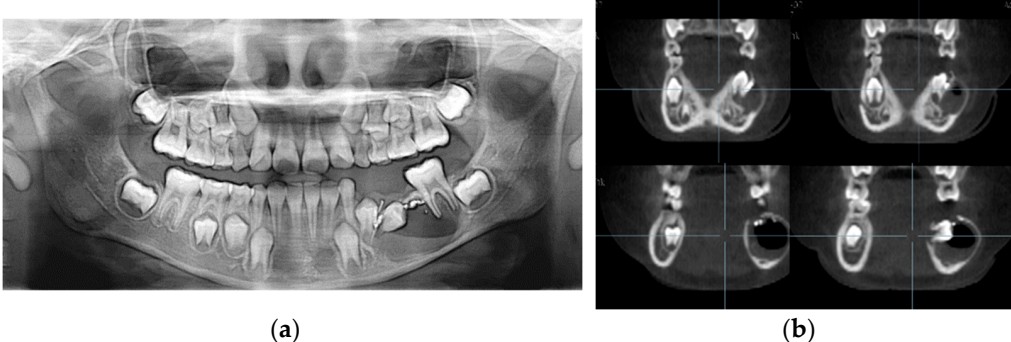

(**a**)  (**b**)

**Figure 4.** Initial visit: (**a**) panoramic radiograph; (**b**) cone-beam computed tomography. The cystic lesion around the successive left mandibular premolars can be seen, and extruded canal filling materials are also observed.

Due to the large size of the cyst, marsupialization to preserve the permanent teeth was planned. Under general anesthesia, marsupialization and incisional biopsy were conducted. The drain was inserted into the cyst cavity and sutured to maintain the openness of the lesion. The biopsy result was a benign cyst with an abscess.

Six months after treatment, the cyst size was significantly reduced, and the premolars had erupted (Figure 5). Thirty months after treatment, the extruded canal filling material remained around the roots of the erupted left mandibular premolars (Figure 6). The patient showed no clinical symptoms.

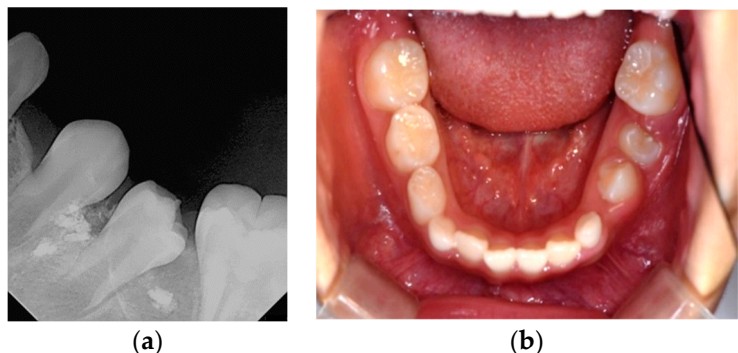

(**a**)  (**b**)

**Figure 5.** Six months after marsupialization: (**a**) periapical radiograph; (**b**) intraoral photo. Left mandibular premolars have erupted and extruded canal filling materials still exist.

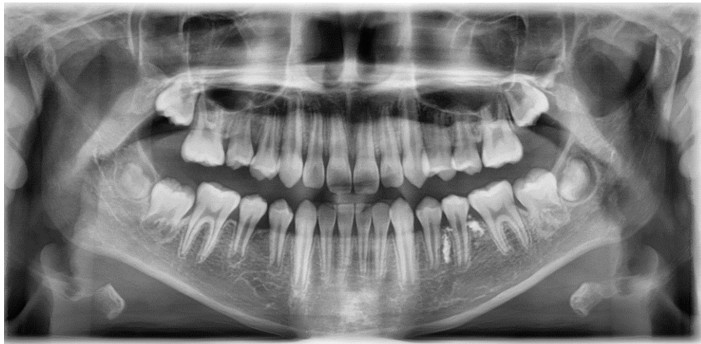

**Figure 6.** Thirty months after marsupialization. Panoramic radiograph. The extruded canal filling materials remain around the root of the erupted left mandibular premolar.

## 2.3. Case 3

An 8-year-old boy visited the Department of Pediatric Dentistry of Kyung Hee University Dental Hospital with facial swelling on the right side of the mandible and pus discharge from the right mandibular primary first molar that had been previously treated at a local dental clinic. His medical history was unremarkable. His dental history was pulpectomy and restoration with a stainless-steel crown on the right mandibular primary first molar a year before at a local dental clinic. According to the guardian's statement, there had been minor, painless swelling in the particular region with fluid discharge after the treatment. On dental examination, pus discharge via the right mandibular primary first molar's disto-lingual sulcus was shown. He had poor oral hygiene and facial swelling on the right side of the mandible. Radiological examination showed a cystic lesion in the successive right mandibular first premolar region with buccal bone expansion and root resorption of the affected primary molar. In addition, a broken file was identified at the distal root tip of the affected primary molar (Figure 7).

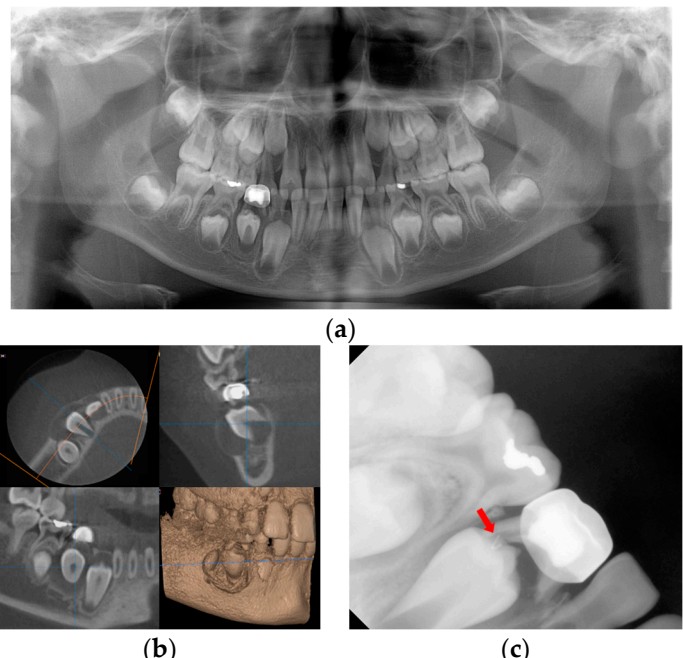

**Figure 7.** Initial visit: (**a**) panoramic radiograph; (**b**) cone-beam computed tomography; the cystic lesion around the successive right mandibular first premolar can be seen and the buccal bone swelling are observed; (**c**) periapical radiograph. A broken file is noted on the distal root tip of the affected primary molar (red arrow).

Under local anesthesia, the right mandibular primary first molar was extracted, marsupialization of the cyst associated with the successive permanent tooth was performed at the second visit, and the separated file was retrieved simultaneously. A removable space maintainer (RSM) with a tube was placed to facilitate the irrigation of the cystic lesion, and the patient's caregiver was instructed to perform saline irrigation twice daily (Figure 8). Postoperatively, analgesics and antibiotics were prescribed for 3 days, and a 0.13% chlorhexidine mouth rinse was prescribed.

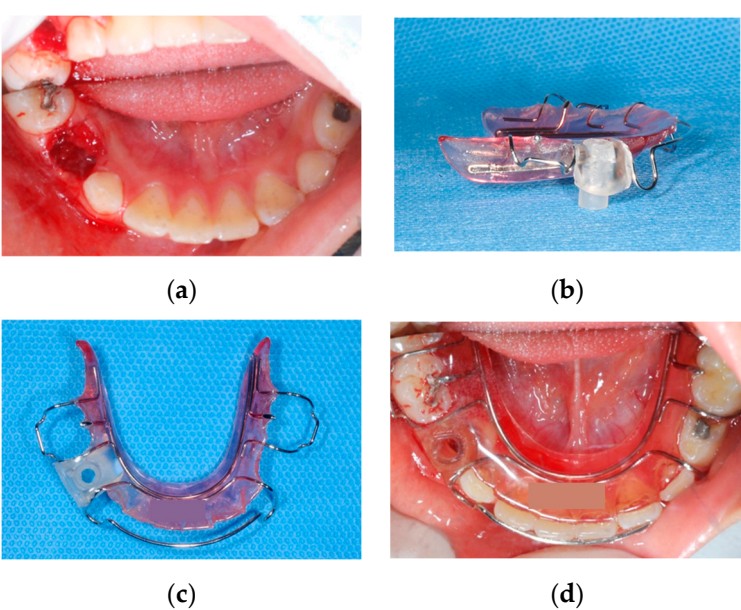

**Figure 8.** The extraction of the right mandibular primary first molar and cyst marsupialization were performed: (**a**) intraoral photo after treatment; (**b**,**c**) removable space maintainer (RSM) with a tube; (**d**) intraoral photo with RSM.

Three days post-treatment, the patient had no complaints, and facial swelling had reduced. Three weeks post-treatment, the facial swelling had nearly subsided, and the RSM was eliminated. Two months post-treatment, the cystic lesion had reduced in size, and the occlusal surface of the mandibular right first premolar was visible intraorally (Figure 9).

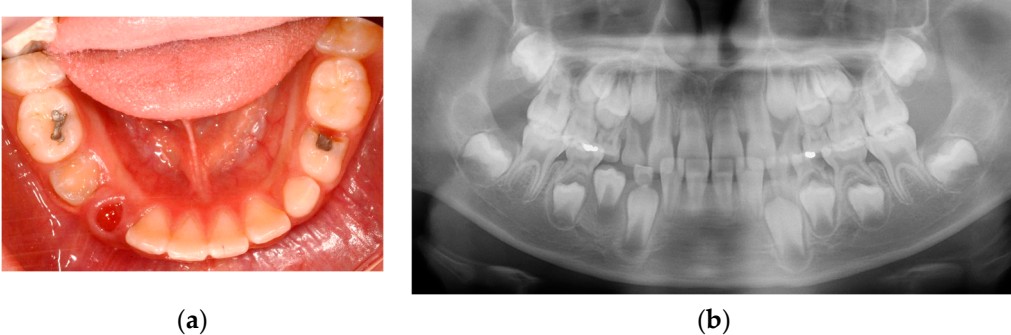

**Figure 9.** Two months after the marsupialization: (**a**) intraoral photo; (**b**) panoramic radiograph.

Four months post-treatment, the bone expansion was remarkably reduced, and dens evaginatus was identified on the premolar. The patient showed no clinical symptoms.

## 3. Discussion

The resorption capacity of the root canal filling material plays an important role in the success of the endodontic treatment of the primary teeth [16]. Severely overextended root canal filling materials typically signify faulty techniques. However, as long as the overextension does not reach the subsequent permanent tooth germs and important structures such as the inferior alveolar nerve or sinuses, or unless the root canal filling materials contain paraformaldehyde, the risk of permanent injury is relatively low [17]. When Vitapex® is extruded into the furcal or apical areas of the primary teeth, it is typically diffused or absorbed by macrophages within 1–2 weeks to as long as 2–3 months, with a success rate of 96 to 100%, and is known to cause no foreign body reactions [12,17–19]. In other studies, the complete resorption of extruded Vitapex® was observed within 6 to 12 months [20].

In cases 1 and 2, it took more than 2 months for the extruded Vitapex® to be resorbed, and especially in case 2, although there were no clinically and radiologically pathologic symptoms, there was no resorption of the extruded filling material after 30 months. This will require continued monitoring afterward until it is fully resorbed. Therefore, it is hard to know exactly how long it will take for the complete resorption of extruded Vitapex®, and overfilling beyond the apex during pulpectomy in the primary teeth should be avoided to protect the successive permanent teeth. Case reports documenting the extrusion of canal filling materials beyond the apex of the primary teeth have been reported in the previous literature [9,10,18,19,21] (Table 1).

**Table 1.** Case reports of filling paste extrusion and file separation in primary teeth.

| Case No. | Authors (Year) | Region Tooth | Patient Age at Accident | Accident | Complications | Follow-Up Treatment | Prognosis (Follow-Up Duration) |
|---|---|---|---|---|---|---|---|
| Case 1 [9] | Nakano et al. (2006) | Left mandibular primary second molar | Unknown | Filling paste (unknown) extrusion | Foreign body reaction due toradiopaque mass, radiolucency from the apex | Observation | Radiopaque masses became smaller but remained detectable and no signs or symptoms around the mass (7 years) |
| | | Right maxillary central incisor | 5-year-old | Vitapex® extrusion | Radiopaque mass superimposed on its permanent successor | Observation | Complete absorption of Vitapex® (18 months) |
| Case 2 [10] | Nurko et al. (2000) | Left and right maxillary central incisor | 17-month-old | Vitapex® extrusion | No complications | Observation | Complete absorption of Vitapex®, clinically asymptomatic (38 months) |
| Case 3 [18] | Nurko et al. (1999) | Left mandibular primary second molar | 2-to-7-year-old (unknown) | Vitapex® extrusion | No complications | Observation | Complete absorption of vitapex®, clinically asymptomatic (14 months) |
| | | Left maxillary primary first molar | | | | | Complete absorption of Vitapex®, clinically asymptomatic (7 months) |
| Case 4 [19] | Bhatia et al. (2012) | Right mandibular primary second molar | 7-year-old | Endoflas extrusion | No complications | Observation | Complete absorption of endoflas (3 months) |
| Case 5 [21] | Chawla et al. (2008) | Primary mandibular molars | 4-to-9-year-old (unknown) | A mixture of zinc oxide powder, calcium hydroxide, and sodium fluoride overfilled | No complications | Observation | Uncomplete absorption of filling materials (2 years) |
| Case 6 | Present case | Left mandibular primary second molar | 3-year-old | Vitapex® extrusion | Buccal sinus tract and gingival swelling, periapical abscess | Extraction | Complete absorption of Vitapex®, clinically asymptomatic, successive tooth is in the pre-eruptive phase, with continuous growth (4 years) |

**Table 1.** *Cont.*

| Case No. | Authors (Year) | Region Tooth | Patient Age at Accident | Accident | Complications | Follow-Up Treatment | Prognosis (Follow-Up Duration) |
|---|---|---|---|---|---|---|---|
| Case 7 | Present case | left mandibular primary molar | 8-year-old | Vitapex® extrusion | Cystic lesion | Marsupialization | Uncomplete absorption of filling materials (30 months) |
| Case 8 [4] | Tulsani et al. (2022) | Right mandibular primary second molar | 6-year-old | File separation | Root resorption, dento-alveolar abscess, pain | Extraction and retrieval of the file, distal shoe space maintainer | Extraction area healing (1 week) |
| Case 9 [6] | Musale et al. (2016) | Right mandibular primary first molar | 7-year-old | File separation | Dento-alveolar abscess, furcal radiolucency | Two-stage pulpectomy and retrieval of the file | Clinically asymptomatic (15 months) |
| Case 10 [7] | Morankar et al. (2020) | Left mandibular primary second molar | 6-year-old | File separation | Spontaneous toothache | Pulpectomy and retrieval of the file | Clinically asymptomatic (unknown) |
| | | | 5-year-old | | | Pulpectomy (failed file retrieval), followed by observation and extraction, band and loop space maintainer | Clinically asymptomatic, normally developing successive premolar (38 months) |
| | | Left mandibular primary first molar | 5-year-old | | Intermittent pain | Extraction (failed file retrieval), band and loop space maintainer | Unknown |
| | | Right mandibular primary second molar | 7-year-old | | | Extraction and retrieval of the file | |
| Case 11 [8] | Mujawar et al. (2016) | Left mandibular primary second molar | 6-year-old | File separation | Root resorption, dento-alveolar abscess, furcal radiolucency | Pulpectomy and retrieval of the file | Clinically asymptomatic (24 months) |
| Case 12 | Present case | Right mandibular primary first molar | 8-year-old | File separation | Facial swelling, pus discharge, cystic lesion, bone expansion, root resorption | Marsupialization, a removable space maintainer with tube | Clinically asymptomatic (4 months) |

Instrument separation or breakage often occurs because of improper use or overuse, such as when an instrument is used multiple times and exceeds its torque limit [22–24]. Separated instruments may lead to endodontic treatment failure when periapical lesions are present [25–27]. Even if the procedure is carried out by experienced surgeons, attempts to remove the fragment could jeopardize the tooth's survival [28]. When file separation occurs in the primary teeth, the retrieval of the separated file is more challenging, time-consuming, and requires more skill than that required for the treatment of permanent teeth. This is

because of the anatomical features of the primary molar canal system and that is requires more care, especially because the retrieval process may affect the subsequent permanent teeth [3]. Therefore, clinicians have to choose between an attempt to retrieve the instrument or extract the tooth followed by space maintenance [28]. Case reports of fractured files during the endodontic treatment of primary teeth have been reported in the previous literature [4,6–8,21] (Table 1).

Factors affecting successive permanent teeth, such as infection, periapical lesions, and file separation, may increase the incidence of developing cysts. Two treatment options are available for cystic lesions: enucleation and marsupialization [29]. Although large radicular cysts are treated by enucleation with the extensive removal of bone and vital teeth, marsupialization can be preferred as a more conservative method to reduce complications [30]. If a cystic lesion is suspected to have invaded a successive permanent tooth, extraction of the primary tooth should be considered for treatment. If possible, subsequent permanent teeth should be preserved with a conservative treatment [30,31]. In case 3, marsupialization was performed as a conservative approach to reduce the cystic lesion. In case 3, file separation occurred in the deciduous molar, and a cystic lesion was associated with the successive teeth. As the successive teeth had not been displaced, marsupialization was performed, and an RSM with a drain was placed to open the way for the decompression of the cyst. A close follow-up should be performed considering the possibility of cystic lesions.

The extrusion of filling materials beyond the root apex and file separation during primary tooth pulpectomy have the potential to cause inflammation and irritate the epithelial rest of the underlying permanent tooth, and can cause the development of an inflammatory cystic lesion in the permanent tooth germ [32–34]. Because these events can have an unfavorable effect, such as underdevelopment of the subsequent permanent tooth, the affected deciduous teeth should be extracted, and marsupialization should be performed on the inflamed area. When an unfortunate accident occurs while treating the primary teeth, treatment options should be decided by considering the least harm to the successive teeth.

## 4. Conclusions

As demonstrated by the three patients in this study and other cases reported in the pertinent literature, complications such as abscess formation and underdevelopment of subsequent permanent teeth can occur in primary molar pulpectomy cases, which may result from iatrogenic incidents. However, with the proper treatment of inflammation, potential removal of foreign bodies, and appropriate follow-up for each case, the impact on the subsequent permanent teeth does not seem serious. Clinicians should be knowledgeable about the anatomical properties of the primary molars, carefully manipulate instruments during pulpectomy, and be aware of the complications that are associated with each treatment. The priority is the preservation of the unerupted permanent tooth. Continuous monitoring of the development and eruption of the subsequent permanent dentition should be performed.

**Author Contributions:** Conceptualization, Y.W.L. and M.S.K.; methodology, K.E.L.; software, Y.K.C.; validation, O.H.N., Y.W.L. and H.-S.L.; formal analysis, H.-S.L.; investigation, O.H.N.; resources, K.E.L.; data curation, Y.W.L.; writing—original draft preparation, Y.W.L.; writing—review and editing, M.S.K.; visualization, S.C.C.; supervision, M.S.K.; project administration, M.S.K. All authors have read and agreed to the published version of the manuscript.

**Funding:** This research was supported by the Basic Science Research Program of the National Research Foundation of Korea, funded by the Ministry of Education, Science, and Technology (No.2021R1G1A1013927).

**Institutional Review Board Statement:** This study was conducted in accordance with the Declaration of Helsinki and approved by the Institutional Review Board of Kyung Hee University Medical Center at Gang-dong (IRB approval no: 2023-07-036).

**Informed Consent Statement:** Written informed consent was obtained from the guardians of all patients who were involved in the study.

**Data Availability Statement:** The data presented in this study are available on request from the corresponding author.

**Conflicts of Interest:** The authors declare no conflict of interest.

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
