# Peer review of "Iatrogenic Incidents in Primary Molar Pulpectomy: A Case Series Report and Literature Review"

_applsci, doi:10.3390/app131911008_

Round 1
Reviewer 1 Report
Dear Authors:
First of all, a cordial greeting
I have carefully read your manuscript, I find it interesting for pediatric endodontists, your review is very good, your table seems appropriate and your images are excellent.
However, there are several points that I would like to comment on:
Its title and purpose establish that errors such as overfilling or fracture of instruments during treatments on primary teeth can affect permanent teeth, thus
Case 4 causes a bit of confusion, you say in the discussion that it could be due to a pulpectomy in the second primary upper molar, however, the first permanent upper molar had an endodontic treatment that began with an occluso-mesial cavity, there is a reasonable doubt that this error could have been made during the treatment of the first upper molar by placing the file towards the interdental space, especially due to the angulation of the fragment of the instrument. One of two options should be: eliminate case 4 or explain all the options that could explain the presence of that fragment in that position.
Another question, did you realize that you had been treated for cases 1 and 2 with Vitapex because you had a reference from the treating dentist? or by the radiographic characteristics of the material?
In the discussion:
It is advisable to discuss the following points:
What you establish in the conclusions between lines 294-298 should be discussed more widely since it is the basis of most of their treatments.
In lines 282 and 283 they say that the instruments should always be eliminated, please provide references to this statement, there are cases in which if it is considered and others, adapting them to pediatric cases due to the divergence of the roots becomes complicated and even dangerous for the tooth survival
Finally, your conclusion must be punctual with respect to what is stated in your title and objective, recommendations and other types of suggestions should go at the end of the discussion.
All of these suggestions are made with all due respect to you as authors and your work.
Reviewer 2 Report
I believe this is a good case report of great clinical significance.
Cases descriptions could be improved. Title is misleading as there are no impact on the successive permanent teeth. Please rephrase title.
Case 1 report: dental history should be better described. How does the operator know the filling material was Vitapex?
Case 2: I don’t believe one can determine the cause of lesion to be the extruded root canal sealer material. Once again, how does one know that is Vitapex? Any previous records stating that was the sealer used?
Case 4 is not relevant to the manuscript as treatment performed and described is on a permanent tooth.
Discussion is poorly written.
Line 278-285: authors are stating to the “best of our knowledge, little is known…” Some literature should be cited in that regards. All citations about fractured instrument are on permanent dentition.
Authors are also stating that overfilling and or instruments fractured in deciduous molars will cause cystic lesions. Please cite literature on that regards
Please review the English. The whole manuscript could be improved in the way it is written.
Round 2
Reviewer 1 Report
Dear authors
I have reviewed your responses and manuscript, I have seen that the reviewers' recommendations were taken into account. I have no further suggestions, the manuscript seems acceptable to me for publication. congratulations
Author Response
We are honored to submit our manuscript to your esteemed journal, and your insightful comments have significantly contributed to enhancing the quality of this manuscript. We wish to express our gratitude once again.
Reviewer 2 Report
Hello, thanks for revising the points I mentioned. I still think some clinical descriptions could be improved, especially in case 1. Just say" temporary restoration" instead of upper temporary filling, for example.
Same with discussion: english can be improved.
Can be improved, especially clinical terms
